

# Compound winter low wind and cold events impacting the French electricity system: observed evolution and role of large-scale circulation

François Collet[1], Margot Bador[1], Julien Boé[1], Laurent Dubus[2], Bénédicte Jourdier[2]

[1]CECI Université de Toulouse, CERFACS/CNRS, Toulouse, France
[2]RTE, Paris, France

Correspondance to : François Collet (collet@cerfacs.fr)

**Abstract.** To reach climate mitigation goals, the share of wind power in the electricity production is going to increase substantially in France. In winter, low wind days are challenging for the electricity system if compounded with cold days that are associated with peak electricity demand. The scope of this study is to characterize the evolution of compound low wind and cold events in winter over the 1950-2022 period in France. Compound events are identified at the daily scale using a bottom-up approach based on two indices that are relevant to the French energy sector, derived from temperature and wind observations. The frequency of compound events shows high interannual variability, with some winters having no event and others having up to 13, and a decrease over the 1950-2022 period. Based on a k-means unsupervised classification technique, four weather types are identified, highlighting the diversity of synoptic situations leading to the occurrence of compound events. The weather type associated with the highest frequency of compound events presents pronounced positive sea-level pressure anomalies over Iceland and negative anomalies west of Portugal, limiting the entrance of the westerlies and inducing a north-easterly flow bringing cold air over France and Europe generally. We further show that the atmospheric circulation and its internal variability are likely to play a role in the observed reduction in cold days, suggesting that this negative trend may not be entirely be driven by anthropogenic forcings. Despite this suggested role for cold days, the observed decrease in compound events does not seem to be strongly influenced by the regional atmospheric circulation.

## 1 Introduction

The transition of the energy system, including the reinforced integration of renewable energy, is necessary to reduce greenhouse gas emissions in accordance with the Paris Agreement. A recent report from the French electricity transmission system operator (Réseau de Transport d'électricité, 2023 ; RTE in the



following) shows that the national energy transition will rely on a widespread electrification of residential
heating, transport, and the industry, along with improving energy efficiency (e.g., thermal renovation of
buildings). Therefore, the electricity demand is projected to increase from 475TWh in 2019 to 580-640 TWh
in 2035, according to scenarios in which France meets its energy transition goals (see scenarios A in RTE,
2023). In light of the future electricity demand, France has expressed its intention to significantly expand its
wind energy capacity in the coming decades. Onshore wind power capacity is projected to increase from
20GW in 2022 to 30-39GW and substantial additional offshore wind farms are also planned, with a total
projected capacity of 18GW by 2035 compared to 0.5GW in 2022 (RTE, 2023).

The production and demand of electricity can be affected by a range of climate conditions over multiple

time scales. In terms of electricity demand during winter, France is known to be one of the most temperature
sensitive among European countries (Bloomfield et al., 2020a). This is mainly explained by the high usage of
electricity for residential heating, which is expected to increase over the next decades (RTE, 2023). Hence,
cold events will likely continue being associated with peak electricity demand based on the projections of the
French future electricity system (RTE, 2023). Besides, part of the electricity production in France relies on
renewable energies that are sensitive to climate conditions including wind speed, solar radiations, and river
flows. As the proportion of renewable energy in the French electricity mix is set to rise, the electricity
production will be more importantly affected by climate variability. In particular, it is anticipated that a higher
proportion of wind power in the electricity mix may lead to higher risks for the production of electricity,
especially during low wind events. This is particularly the case in winter, when solar generation represent a
smaller share of the electricity production (Grams et al., 2017; Otero et al., 2022b). Hence, in France, it can
be challenging to ensure adequate electricity supply and demand due to the occurrence of multivariate
compound events (Zscheischler et al., 2020), such as low wind and cold events, which can create stressful
situations. The aim of this study is to characterize compound low wind and cold events in France.

Overall, there is little information in the literature on the observed evolution of compound low wind

and cold events in France and Europe. A body of studies focuses on related events using electricity supply and
production data. For instance, an electricity supply drought is defined by a sequence of days with low
renewable electricity production and high electricity demand (Raynaud et al., 2018). Most of these studies
focus on the characterization of the statistical properties of these events (Otero et al., 2022a, b; Raynaud et al.,
2018; Tedesco et al., 2023) or their drivers (Bloomfield et al., 2020a; Ravestein et al., 2018; Thornton et al.,
2017; van der Wiel et al., 2019a, b). Only a limited number of these studies focus on their temporal evolution
in the context of climate change. Van der Wiel et al. (2019a) show that the frequency of electricity supply
droughts in Europe is reduced in a 2°C warmer world compared to present day conditions, using projections
from two global climate models. Although there is a gap in the understanding of the past evolution of
compound low wind and cold events, changes in low wind or cold events have been investigated
independently. Rapella et al. (2023) showed that the number of low wind events decreases in the ERA5
reanalysis over the 1950-2022 period. However, they focus only on offshore regions such as the Bay of Biscay,





the North Sea, and the Channel, in summer and at the annual scale. Focusing on cold temperature conditions
in winter, there is evidence that the frequency and intensity of cold spells have decreased over the last decades
in Europe (Cattiaux et al., 2010; Seneviratne et al., 2021; Van Oldenborgh et al., 2019). While there is clear
evidence that climate change leads to a reduction in cold events, there are still major uncertainties regarding
low wind events. It is therefore difficult to anticipate how compound low wind and cold events may change
in the coming decades as there is a lack of understanding of their past evolution. An objective of this study is
to assess the evolution of these compound events in the observational record.
This work also focuses on the influence of the regional atmospheric circulation on the occurrence and
evolution of compound low wind and cold events. The atmospheric circulation is an important driver of
temperature variability (Plaut and Simonnet, 2001) and wind speed variability (Najac et al., 2009) in France,
and here we aim to further assess its influence on compound events in winter. In the literature, different
approaches have been used to explore the influence of the atmospheric circulation and its variability in
favoring particular meteorological situations that affect the electricity sector. This includes identifying weather
regimes of interest (Otero et al., 2022b; van der Wiel et al., 2019b; Tedesco et al., 2023), targeted circulation
types (Bloomfield et al., 2020b), and circulation regimes based on large-scale conditions leading to critical
situations for the electricity system such as days with extremely high electricity demand (Thornton et al.,
2017). Tedesco et al. (2023) showed that compound low wind and cold events in France are mostly associated
with positive anomalies of geopotential height at 500hPa over Iceland and negative anomalies over the Azores.
Otero et al. (2022b) showed that situations of limited production of electricity from wind and solar energies
co-occurring with cold events are mostly associated with positive anomalies of geopotential height at 500hPa
over the North Sea region.
Finally, we investigate to what extent the regional atmospheric circulation and its variability contribute
to the past evolution of compound low wind and cold events in France. Several studies found that recent
changes in the large-scale circulation play a role in the winter trend in mean temperature across Europe (Deser
and Phillips, 2023; Sippel et al., 2020; Saffioti et al., 2016), and in the decreasing occurrence and intensity of
cold extremes (Horton et al., 2015; Terray, 2021). Using a dynamical adjustment approach based on
observation data (Terray, 2021), we explore the role of the changes in atmospheric circulation in the observed
trend in compound low wind and cold events in France.
This paper is organized as follows: section 2 presents the data and the method used, section 3 presents
the main results and section 4 includes a conclusion and discussion of the findings.

## 2 Data and Method

First, the data and the methodology used to identify low wind events, cold events, and compound low
wind and cold events responsible for stressful situations for the adequacy between electricity demand and
supply in France are described.





### 2.1 Observations and reanalyses of atmospheric variables

The ERA5 reanalysis data (Hersbach et al., 2020) is used over the period 1950-2022. ERA5 is available on a regular grid with a resolution of about 30 km in Europe. In particular, the hourly wind speed (at 100 m) and the daily near-surface air temperature (at 2m) are used for the calculation of the wind capacity factor and temperature indices, respectively (section 2.3 and 2.4). Daily mean sea level pressure is also used for classification of the large-scale circulation into weather types (see section 2.6) and dynamical adjustment (section 2.7). In addition to the ERA5 reanalysis, wind and temperature data from the MERRA-2 reanalysis (Gelaro et al., 2017) are considered. MERRA-2 is available at a horizontal resolution of about 60 km over Europe, over the 1980-2022 period. Hourly near-surface air temperature and wind (at 50 m) are used. We also consider in situ temperature observations from the gridded E-OBS dataset (Cornes et al., 2018) over the 1950-2022 period, available on a regular grid with a horizontal resolution of about 30 km in Europe.

This study is mainly focused on an extended winter period, from November to March, when compound low wind and cold events occur in France. By convention, hereafter, winter 1951 corresponds to the period from November 1950 to February 1951 and so one.

### 2.2 Observations of the wind power production and electricity demand in France

Hourly observed data for the wind power production and electricity demand in France are taken from the éCO2mix dataset (https://odre.opendatasoft.com/explore/dataset/eco2mix-national-cons-def/information/?disjunctive.nature), over the 2012-2020 period. The French wind power installed capacity is available at 3-monthly time intervals over the 2012-2020 period at https://www.statistiques.developpement-durable.gouv.fr/publicationweb/549. Hourly observed wind capacity factor is calculated using the hourly observed wind power production from éCO2mix, which is divided by the wind power installed capacity in France of the corresponding 3-monthly interval.

### 2.3 Wind capacity factor index

Several studies (Bloomfield et al., 2022; Jourdier, 2020; Olauson, 2018; Staffell and Pfenninger, 2016) demonstrated that it is possible to calculate hourly wind capacity factor at country-scale with a good accuracy using wind speed from reanalysis data in Europe. Here, we use a similar approach to calculate the French wind capacity factor index over the 1951-2022 period.

This approach requires information at each wind farm site, which are taken from The Wind Power database (https://www.thewindpower.net/), including the location, rated power, hub height, and power curves at each site (upon availability). Only wind farms operational in 2021 are used (i.e., those with "in production" status). This represents a total number of 1661 wind farms and a total installed capacity of 19GW. Wind farms and related wind power installed capacity are concentrated in the North-East of France (Figure 1a). While the installed wind power capacity is fairly accounted for in this database, there is a substantial amount of missing





data regarding the hub heights and the power curves (~29% and ~7% of wind farms, respectively). Missing
data is filled in following the methodology introduced in Jourdier (2020), which broadly consists in taking
characteristics from wind farms identified as similar in terms of rated power, rated diameter, rated wind speed,
cut-in and cut-off wind speed.
Calculation of the wind capacity factor first requires interpolating ERA5 hourly wind speed from 100
m at each wind farm's hub height. This is done using a power law ($\alpha$=0.14; Manwell, 2010; van der Wiel et
al., 2019a). Then, using the power curve of each wind farm, wind speed at the hub height is converted into
power production. Finally, the hourly wind capacity factor over France is estimated by summing the power
production from all wind farms, and dividing this total power production by the total installed capacity.
Finally, hourly wind capacity factors are averaged to daily values to further identify low wind days (section

2.5).

The daily wind capacity factor index computed with this approach is extremely well correlated with
observations over their 9 common winters (r=0.99, Figure 2a), highlighting the relevance of using ERA5 data
in this context.
**2.4 Temperature index representative of the demand in electricity**
The temperature index is defined following an approach used operationally by RTE that consists in
calculating a weighted average of temperature data from 32 cities in France (Figure 1.b), which is
representative of the electricity demand in France. First, the near-surface air temperature in ERA5 at the grid-
cell closest to each city location is selected. Then, temperatures are corrected based on the difference between
the elevation of the grid cell and the elevation of the in situ station for each city, assuming a vertical gradient
of temperature of -6.5°C/km. Finally, the weighted average of temperature at the 32 locations is calculated
over the 1950-2022 period.
A strong anti-correlation of -0.82 is found between the temperature index and the observed electricity
demand in winter (Figure 2b). This highlights the relevance of the temperature index as a proxy for the French
demand in electricity.





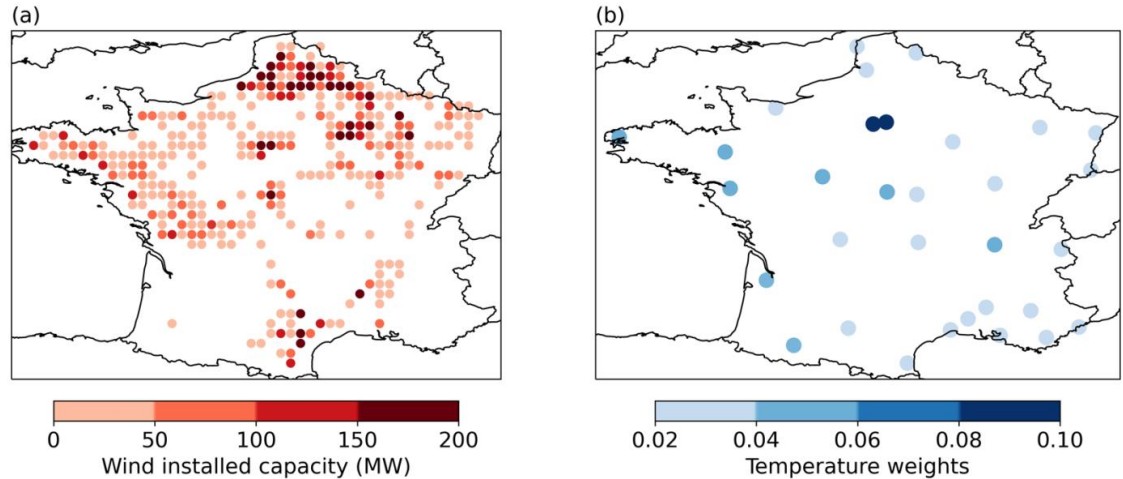

Figure 1: (a) Spatial distribution of the wind power installed capacity (MW) in France in 2021 from the WindPower.net dataset used for the calculation of the wind capacity factor index. (b) Location of the 32 French cities and associated weights (no unit) used for the calculation of the temperature index.

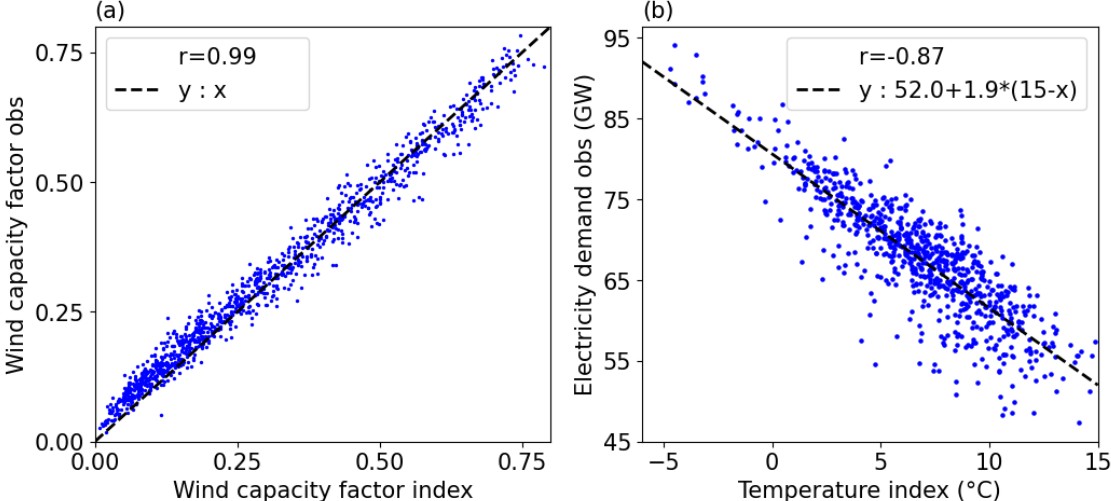

Figure 2: (a) National French wind capacity factor index as calculated with ERA5 (no unit; X-axis) versus observations (no unit; Y-axis) in winter over the 2012-2020 period. The correlation coefficient is given in the top left corner, and the black dotted line represents the y:x function. (b) Temperature index as calculated in ERA5 (°C; X-axis) versus observations of the electricity demand (GW; Y-axis) in winter over the 2012-2020 period, excluding week-ends and bank holidays. The correlation coefficient is given in the top right corner. The linear regression line between the temperature index and the electricity demand observations is shown by the black dashed line. The corresponding linear regression equation, in the form y=y(15°C)+a*(15°C-x), where 15°C is the threshold of residential heating and a the thermosensitivity of the electricity demand, is shown in the top right corner.





**2.5 Identification of low-wind days, cold days and associated compound events**

In this study, compound low wind and cold events are defined as days when cold temperature and low wind conditions co-occur (green points in Figure 3). Cold days are defined here as days with the temperature index below 0°C, corresponding to the 5th percentile of the distribution of the temperature index in winter (blue points in Figure 3). Low wind days (red points in Figure 3) are defined as days with a wind capacity factor index below a certain threshold (here the 23th percentile), which corresponds to a wind capacity factor of 0.15 in the distribution of observations in winter.

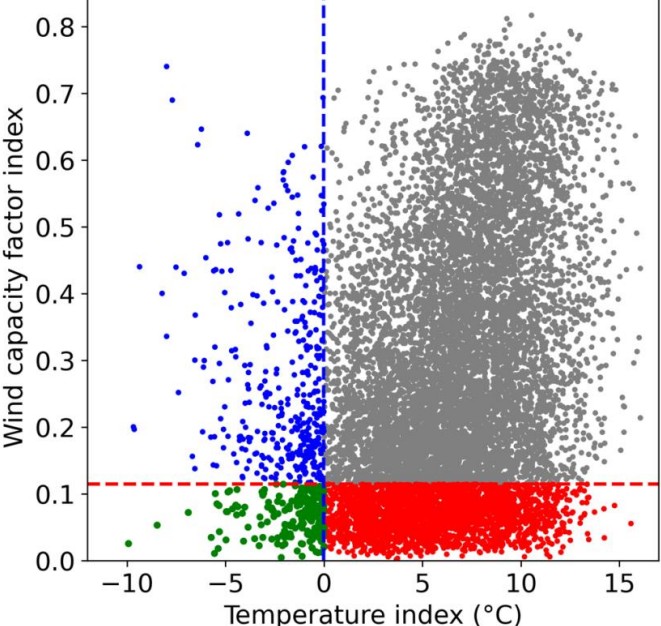

Figure 3: Wind capacity factor index (no units; Y-axis) and temperature index (°C; X-axis) calculated with ERA5 for each winter day over the 1951-2022 period. Red and blue dashed lines show the thresholds used to identify low wind days and cold days (red and blue points, respectively). Compound low wind and cold events are identified by the green dots.

**2.6 Classification into weather types**

A classification of mean sea-level pressure fields on low wind days is conducted using the k-means unsupervised classification method (e.g., Cassou, 2008; Falkena et al., 2020). This allows classifying daily synoptic conditions into different large-scale atmospheric circulation types, or weather types. Here, low wind days solely are considered for the classification instead of compound low wind and cold events because the corresponding sample size is larger (2549 days compared to 182 days, respectively; see Figure 3, Figure 4b,





and further discussions in section 3). However, in a second phase, we assess how cold days and therefore
compound events are distributed across the different weather types leading to low wind days.

This classification algorithm is first applied repeatedly for different domains and number of clusters.
The objective is to minimize locally the ratio of intra-type to inter-type variance of the temperature index,
while keeping a reasonable number of weather types. Thanks to this procedure, the classification of low wind
days that allows for the best differentiation of the temperature index is chosen. This procedure leads to a
domain whose limits are [30°W-30°E/33°S-70°N], which covers the North-Western Europe region, and a total
number of four clusters.

**2.7 Dynamical adjustment**

The main objective of dynamical adjustment is to derive an estimate of the contribution of atmospheric
circulation to the variations of a variable of interest (Terray, 2021; Deser et al., 2016; Sippel et al., 2019). In
this study, we use dynamical adjustment to estimate the contribution of atmospheric circulation to the
variations of cold days, low wind days, and compound events. Hereafter, the contribution of atmospheric
circulation is referred to as the dynamic component.

First, we estimate the dynamic component of the wind capacity factor and temperature indices. To that
purpose, the constructed analogue approach is used (Terray, 2021; Boé et al., 2023; Deser et al., 2016).
Following Lorenz (1969), analogues are defined as days with very similar atmospheric circulation. As finding
genuinely good analogues in a finite database could be difficult, synthetic analogues can be constructed
through the linear combination of the atmospheric circulation corresponding to a large number of more or less
good analogues (Van Den Dool, 1994).

First, for each target day of the winters 1951-2022, the 1500 closest analogues are searched in winter
using the Euclidean distance calculated with ERA5 mean sea-level pressure interpolated on a 2°x2° grid on
the North-Western Europe domain (section 2.6). The winter of the target day is excluded from the search pool.
Then for each target day, a subset of 1300 analogues are randomly selected from the 1500 analogues, and the
optimal linear combination of this subset of 1300 analogues that best matches the mean seal level pressure of
the target day is calculated. This allows obtaining a constructed analogue for the target day. This procedure is
repeated 50 times, to obtain 50 constructed analogues for each target day and the corresponding 50 sets of
optimal weights. While the 50 constructed analogues of each target day have very similar atmospheric
circulation to the target day, this procedure, together with the large number of analogues used allows us to
sample different land surface and ocean conditions that might otherwise influence the estimate of the dynamic
components (Terray, 2021).

For each target day, the wind capacity factor and the temperature indices are then reconstructed by
applying the same set of optimal linear weights to the corresponding wind capacity factor index and detrended
anomalies of the temperature index, respectively. There are 50 reconstructions of the wind capacity factor and
the temperature index per day over the winters 1951-2022. As we are interested in separating the trend due to




large-scale circulation from thermodynamically-forced changes, an estimate of the forced trend of the
temperature index anomaly for each winter month is removed before applying the dynamical adjustment. This
low-frequency trend is estimated using a low-frequency LOESS smoother as done in Terray (2021). Finally,
a best estimate of the dynamic component of the wind capacity factor index and the temperature index are
derived by averaging the 50 reconstructions of the wind capacity factor index and the temperature index,
respectively.
Finally, the dynamic component of low wind days and cold days is defined using the same thresholds
as for the definition of cold days and low wind days (i.e., the 5th percentile and the 23th percentile,
respectively; section 2.5). This allows the dynamic component of compound events to be identified as days
when both the dynamic component of low wind days and cold days occur.
**3. Results**
**3.1 Climatological characteristics and observed evolution of compound low wind and cold events**

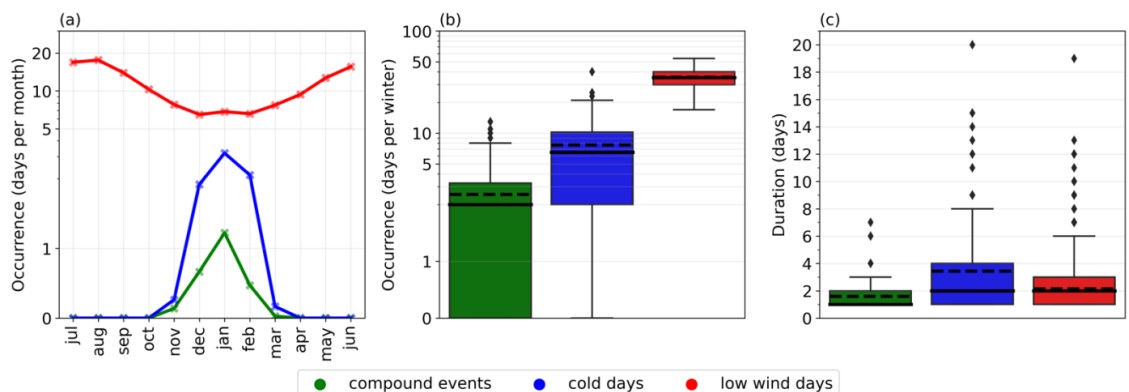


Figure 4: (a) Monthly mean number of compound low wind and cold events (green), cold days (blue), and low
wind days (red); Distributions of (b) the number of days per winter and (c) duration of compound low wind
and cold events, cold days, and low wind days in winter over the 1951-2022 period in ERA5. The solid line
and the dashed line in the boxplots in (b) and (c) show the median and the average, respectively.

There is a clear seasonality in the occurrence of compound events, which are concentrated in winter
(November to March; Figure 4a). This is well explained by the seasonality of cold days, which occur from
November to March in France. Conversely, low wind days are less frequent in winter, with an average of 7
days for winter months compared to an average of 13 days for other months.
The median number of compound events per winter (2 days; Figure 4b) is a third of the median number
of cold days per winter (6 days; Figure 4b). The median number of low wind days per winter reaches 35 days,
and is therefore substantially higher than for compound events and cold days. In terms of year-to-year
variability, we find that the number of compound events ranges from 0 to 13 days per winter, while there are





from 0 to 40 cold days and 17 to 54 low wind days. When compared to the mean, the interannual variability
is thus higher for the occurrence of compound events and cold days compared to low wind days.
On average in winter, the duration of compound events is estimated to be around 2 consecutive days, 3
days for cold days and 2 days for low wind days (Figure 4c). The maximum duration of compound events is
7 consecutive days, corresponding to the period between 17 and 23 January 1987, at the end of a severe 13-
day cold spell. Overall, compound low wind and cold events are relatively rare and generally short-lived, but
they can last for a few days and up to a week occasionally.

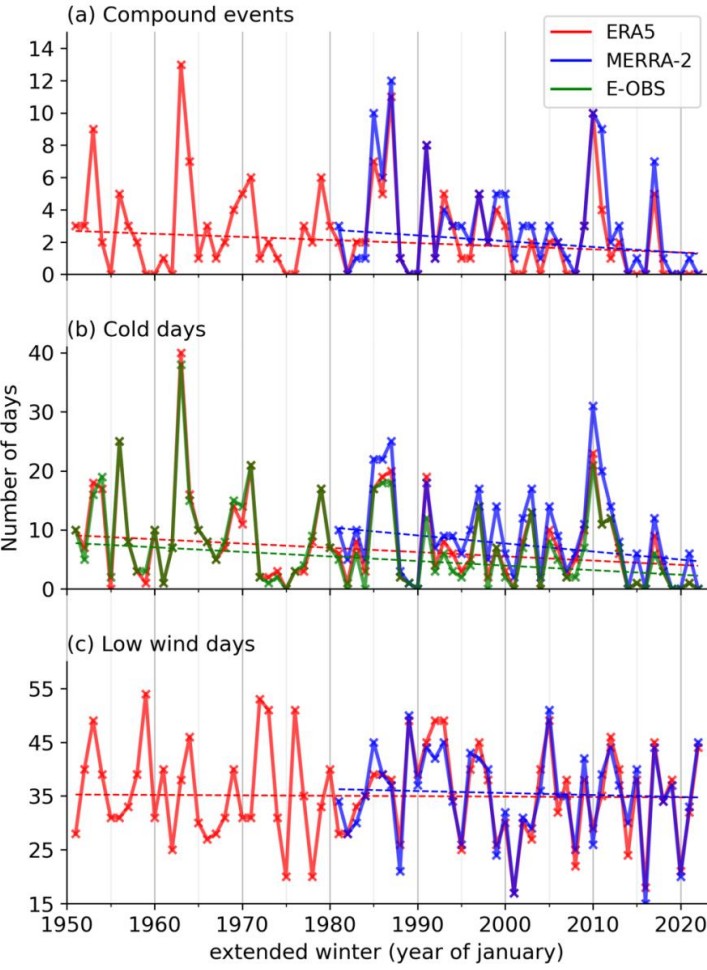


Figure 5: Interannual evolution of the number of (a) compound low wind and cold events, (b) cold days, (c)
and low wind days per winter in ERA5 (in red; 1951-2022), MERRA2 (in blue; 1981-2022) and EOBS (in
green; 1951-2022) datasets. Dashed lines show the linear trend (calculated with the Theil-Sen estimator; see
Table 1 for the slope value and associated significance).




| Data | ERA5 | | MERRA-2 | | E-OBS | |
|---|---|---|---|---|---|---|
| Time period | 1951-2022 | 1981-2022 | 1951-2022 | 1981-2022 | 1951-2022 | 1981-2022 |
| Compound events | -0.19 (0.02) | -0.43 (0.01) | / | -0.36 (0.1) | / | / |
| Cold days | -0.72 (0.02) | -1.03 (0.08) | / | -1.36 (0.08) | -0.78 (0.0) | -0.67 (0.16) |
| Low wind days | -0.08 (0.59) | -0.45 (0.48) | / | -0.37 (0.72) | / | / |

Table 1: Trend (slope in days/decade) and associated p-value, in the number of compound low wind and cold events, cold days, and low wind days in ERA5, MERRA-2 and E-OBS over their respective time period (as indicated in the first row). The slope is calculated with Theil-Sen estimator and the p-value with the Mann-Kendall test. Significant trends with p<0.05 are highlighted with grey shading. Empty cells correspond to missing data.

Further looking into the year-to-year differences in the number of compound low wind and cold events, we find substantial interannual variability (Figure 5a). Some winters stand out as extreme cases, such as 1963, 1985, 1987, and 2010. In particular, the exceptional winter 1963, is the most extreme winter with 13 days of compound events (Figure 5b). Winter 1963 is the coldest winter ever recorded over Western Europe (Hirschi and Sinha, 2007) and our results further show that low wind days were co-occurring for some of its cold days. Overall, there is a good agreement between ERA5 and MERRA-2 over the shorter 1981-2022 period. This includes the characterization of the most extreme winters in terms of compound events, although MERRA2 generally shows a slightly higher number of compound events per winter.

The interannual variability of compound events is primarily driven by the variability of cold days compared to the variability of low wind days (r=0.86 and r=0.19 in ERA5, respectively; Figure 5a,b). In particular, the highest numbers of compound events are found in years also characterized by the highest numbers of cold days, but not necessarily in years with the highest numbers of low wind days (e.g., 1963, 1987, 2010, Figure 5a,b,c). This might be related to the larger sample of low wind days per winter on average compared to the number of cold days, as defined in this study (section 2.5).

Over the 1951-2022 period, there is a significant decrease in the number of compound events per winter in ERA5 (-0.19 days per decade; Figure 5a and Table 1). Over the shorter period in common between ERA5 and MERRA2, compound events have also decreased significantly in ERA5, and at a higher rate (-0.43 days per decade). MERRA-2 shows a slightly weaker decrease in compound events (-0.36 days per decade) compared to ERA5, which is not significant at the 0.05 level (p=0.10). In terms of low wind days, no trend is





detected in ERA5 over both the longer and shorter periods, and both reanalyses agree on the absence of a
trend. Conversely, cold days have significantly decreased over the longer period in both the ERA5 reanalysis
and the E-OBS observations, and at a similar rate of -0.72 and -0.78 days per decade (respectively; Figure 5b
and Table 1). Interestingly, over the shorter period in common with ERA5, MERRA2 and EOBS, the
significance of the negative trend is lost, suggesting that this period might be too short for the influence of
anthropogenic forcings to emerge from internal variability, contrary to what is observed on the longer period.

### 3.2 Role of large-scale circulation.

On average, the synoptic conditions leading to the occurrence of compound low wind and cold events
are characterized by strong positive mean sea-level pressure anomalies over the British Isles and relatively
less intense negative anomalies centred on the Azores (Figure 6a). Overall, the average large-scale circulation
during compound events is very well spatially correlated with that of cold days (Figure 6b), but the intensity
of the positive anomalies and associated pressure dipole are weaker in the case of compound events. The
anomalies in mean sea-level pressure are somehow different during low wind days compared to compound
and cold events. Positive sea level pressure anomalies are found further south over the North Sea, with
relatively lower intensity, and the negative anomalies over the Azores are not as clear (Figure 6c).
By definition, during compound low wind and cold events, France experiences calm and cold
temperature conditions. On average during compound events, the negative anomalies of wind speed and
temperature expand over a wider domain, comprising Germany and the British Isles, with anomalies up to -
40% and -7.5°C, respectively (Figure 6d,g). The negative temperature anomalies over France and surrounding
countries are slightly weaker during compound events compared to cold days (Figure 6g,h). These cold
anomalies are induced by a north-easterly flow advecting cold polar air towards western Europe. During cold
days, and compared to compound events, the negative anomalies in wind speed are less intense, the advection
of cold air is stronger, and thus colder temperatures are experienced over western Europe. During low wind
days, negative wind anomalies are found over western Europe, with intensities rather similar to those during
compound events, along with neutral temperature anomalies (Figure 6f,i). It is important to acknowledge that
these average climate conditions might hide a variety of different regional atmospheric circulations, further
explored in the following using a weather type analysis.




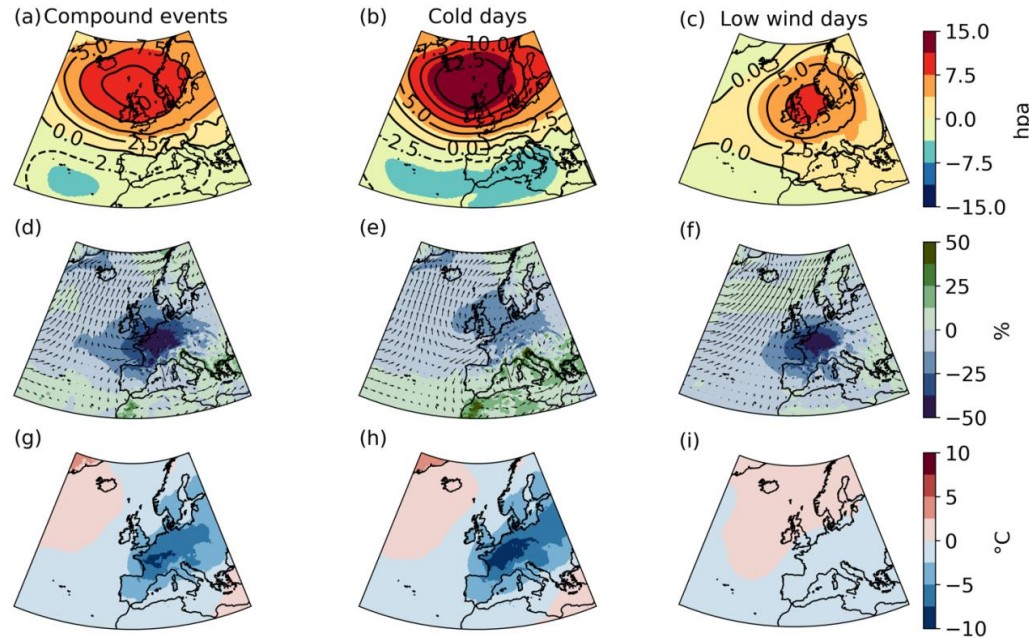

Figure 6: Composite of (a,b,c) sea-level pressure anomalies (hPa) with solid and dashed contours corresponding to positive and negative anomalies respectively, (d,e,f) 100 m wind speed relative anomalies (% of climatological mean; shadings) and wind direction (arrow), and (g,h,i) near-surface air temperature anomalies, in average during (a,d,g) compound low wind and cold events, (b,e,h) cold days, and (c,f,i) low wind days. The anomalies are calculated over the 1951-2022 period in ERA5.

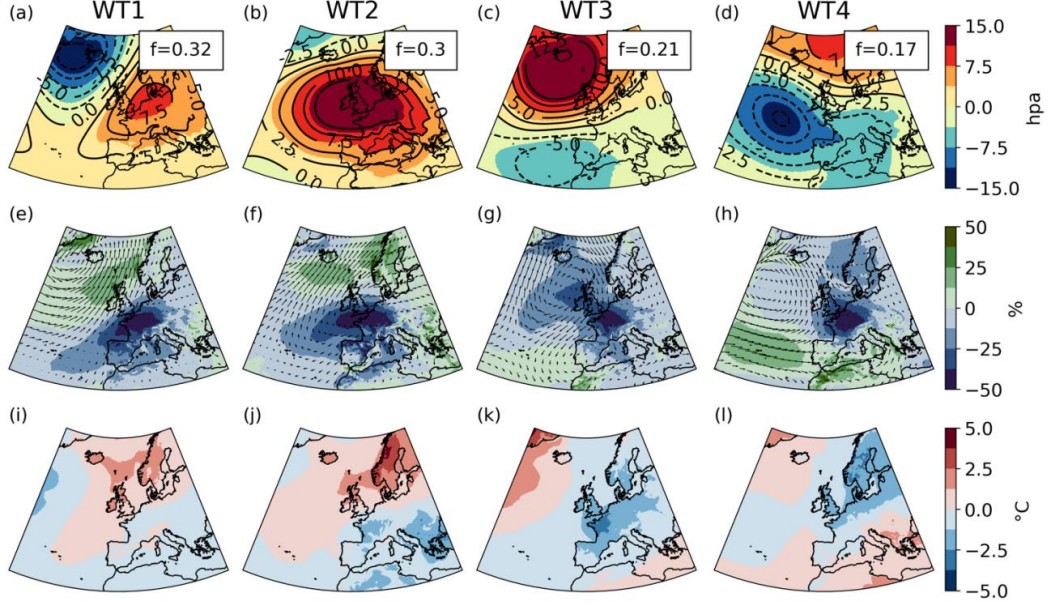





Figure 7: Composite of (a,b,c,d) sea-level pressure anomalies (hPa) with solid and dashed contours
corresponding to positive and negative anomalies respectively, (e,f,g,h) 100 m wind speed relative anomalies
magnitude (% of climatological mean; shadings) and wind direction (arrows), and (i,j,k,l) near-surface air
temperature anomalies corresponding to the weather types (a,e,i) WT1, (b,f,j) WT2, (c,g,k) WT3 and (d,h,l)
WT4. The anomalies are calculated over the 1951-2022 period in ERA5. The frequency (f) of the weather
types is shown in the upper right corner in panels a,b,c,d.

The four weather types obtained with the k-means algorithm (section 2.6) help to identify the most
favorable synoptic situations leading to the occurrence of compound low wind and cold events in France, and
over western Europe more generally.
The frequency of weather types is rather similar, and ranges from 0.17 (WT4) to 0.32 (WT1). While all
four weather types are characterized by low wind and cold temperatures in France (by definition), they reveal
a diversity of regional atmospheric conditions (Figure 7):
• WT1 is characterized by positive sea-level pressure anomalies over the Netherlands and northern
Germany, and negative anomalies over Iceland. The positive anomalies block the entry of the
westerlies at the western border of Europe and deviate them further north, thus advecting relatively
warm and humid air over northern Europe, and inducing a substantial decrease in wind speed along
with cold anomalies in France and western Europe.

• WT2 shares blocking-like characteristics with WT1, but with more intense positive sea level
pressure anomalies and over a wider domain extending further west, pushing the negative SLP
anomalies further to the north-west corner of the domain. As in WT1, the westerlies are derived
north of Europe, inducing a similar dipole of warmer temperatures in the north and colder
temperatures under the positive pressure anomalies. In France and southern Europe in general, and
compared to WT1, the negative anomalies in wind and temperature are enhanced because of the
amplified positive pressure anomalies.

• WT3 shows pronounced positive sea-level pressure anomalies over Iceland and negative anomalies
west of Portugal. This WT resembles the most to the average atmospheric conditions during
compound events (Figure 6a). The dipole of pressure anomalies results in a strong north to north-
easterly flow advecting cold air masses from Scandinavia to France. This weather type is associated
with the coldest temperatures over France compared to the other weather types, and generally over
the entire European domain that also experiences low wind conditions.

• WT4 is rather different from WT1, WT2 and WT3 as it is characterized by substantial negative
sea-level pressure anomalies in the eastern Atlantic and positive anomalies over the Norwegian
Sea. These pressure anomalies induce low wind conditions in France and generally the northern
part of Europe, and a reinforcement of the westerlies in the southern part of the domain. This is



associated with colder temperatures in the north, including the northern part of France, and positive
or low temperature anomalies in south-western Europe.

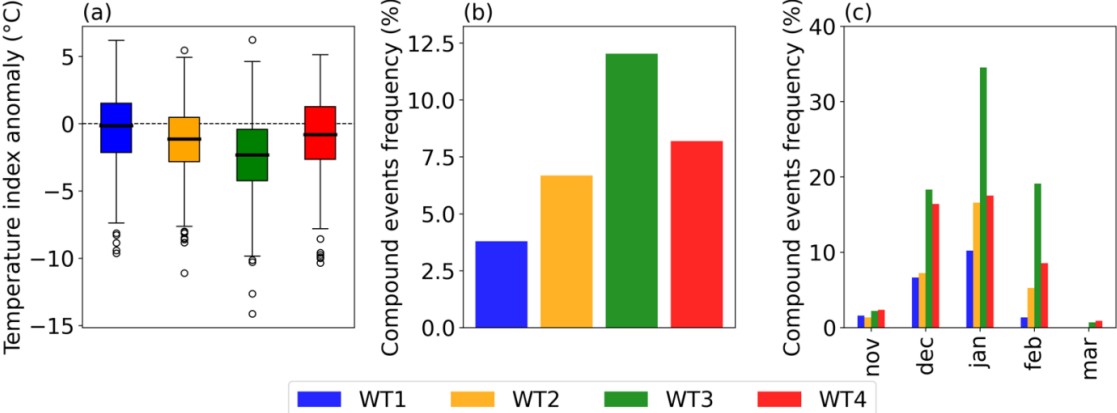


Figure 8: (a) Distribution of temperature index anomalies for each weather type (WT; as defined in Figure 7
and indicated in inserted legend); (b) Frequency of compound low wind and cold events for each weather type
(in % of the weather type size). (c) Frequency of compound low wind and cold events for each weather type
and each individual winter month (in % of the weather type size for a given month). The anomalies and
frequencies are calculated over the 1951-2022 period in ERA5.

The temperature index shows a substantial intra-type and inter-type variability (Figure 8a). WT3 is the
coldest weather type over France overall, with the lowest median of the temperature index. Yet, all weather
types present very cold days, with anomalies as large as -10°C for WT1 and WT4, and -14°C for WT3. The
frequency of compound events when a particular weather type occurs varies from 4% in WT1 to 12% in WT3,
while WT2 and WT4 present similar values of 7% and 8% (Figure 8b). Importantly, the weather type WT3,
which is associated with the highest frequency of compound events, also leads to negative anomalies in wind
speed and temperature across Europe (Figure 7c,g,k). It suggests that this weather type might challenge the
electricity system on a larger scale than just in France.
The frequency of compound events in each weather types shows important monthly variations. For all
weather types, the frequency of compound events is higher in January, when climatological temperature
reaches the lowest values, compared to other months (Figure 8c). This is especially the case for WT3, for
which nearly 35% of days occurring in January are compound events. This important role of the temperature
seasonality within each weather type is consistent with the overall seasonality of compound events discussed
in section 3.1.






| WT1 | WT2 | WT3 | WT4 |
|------|------|------|------|
| 0.0 (0.78) | 0.56 (0.16) | -0.27 (0.29) | -0.59 (0.01) |


Table 2: Trend (slope in days/decade) and associated p-value in the frequency of each weather type (WT; as
defined in Figure 7) in winter over the 1951-2022 period in ERA5. The slope is calculated with the Theil-Sen
estimator and the p-value is calculated with the Mann-Kendall test. Significant trends with $p<0.05$ are
highlighted with grey shading.

Only the frequency of WT4 shows a significant negative trend over the 1951-2022 period (-0.59 day per
decade, p=0.01; Table 2). The frequency of WT2 is found to increase (+0.56 day per decade), whereas WT3,
which is associated with the highest frequency of compound events, decreases (-0.27 day per decade) over the
observed period. Trends for both WT2 and WT3 are however not significant.
To estimate the contribution of the trends in weather type frequencies on the overall evolution of
compound events, the trends are multiplied by the frequency of compound events for the corresponding
weather type, as done in Horton et al. (2015). Then, the respective contributions from all four weather types
are added to estimate the overall influence of the trends in weather type frequencies. Overall, the trends in
weather type frequencies lead to a weak decrease in the frequency of compound events of 20%. This analysis
suggests a relatively minor influence of large-scale circulation on the trend of compound events. However,
due to significant intra-type variability, a simple change in the frequency of a few weather types may not
capture the full range of circulation changes.





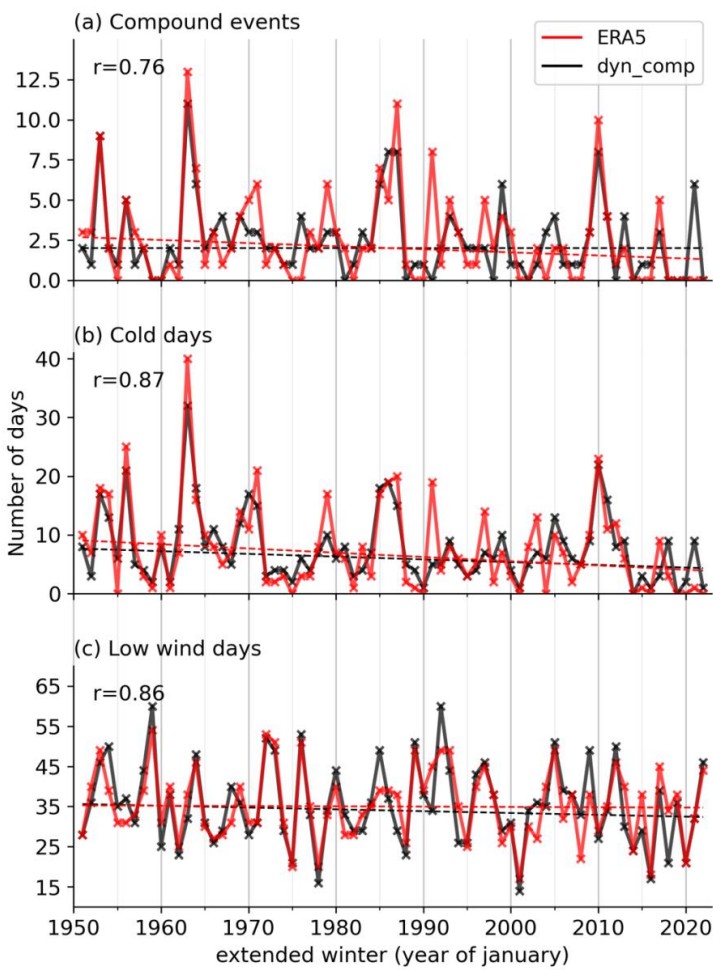


Figure 9: Interannual evolution of the number of (a) compound low wind and cold events, (b) cold days, and (c) low wind days in winter over the 1951-2022 period in ERA5 (red) and in their respective dynamic components (i.e. dyn_comp, in black). For each event, the correlation coefficient between ERA5 and their respective dynamic components is shown in the upper left. Dashed lines show the linear trend (calculated using the Theil-Sen estimator; see Table 3 for the slope value and associated p-value).

|  | Compound events | Cold days | Low wind days |
|---|---|---|---|
| ERA5 | -0.19 (0.02) | -0.72 (0.02) | -0.08 (0.59) |
| Dynamical component | 0.0 (0.10) | -0.47 (0.08) | -0.45 (0.44) |


Table 3: (first row) Trend (slope in days/decade, and associated p-value) in the frequency of low wind days,
cold days, and compound low wind and cold events in winter over the period 1951-2022 in ERA5 (first row;



same trend estimates as in Table 1) and in their respective dynamic component (second row; section 2.7). The
slope is calculated with the Theil-Sen estimator and the p-value is calculated with the Mann-Kendall test.
Significant trends with p<0.05 are highlighted with grey shading.

The dynamical adjustment approach described in section 2.7 is now used to better quantify the role of the
large-scale circulation in the evolution of compound low wind and cold events. The interannual variability in
the occurrence of both cold days and low wind days is very well explained by the large-scale circulation
(correlations with the corresponding dynamic component of 0.87 and 0.86, respectively; Figure 9a,b).
Therefore, the interannual variability in the number of compound events is also well explained by the large-
scale circulation (correlations with the corresponding dynamic component of r=0.76, Figure 9a). Extreme
winters in terms of compound events such as 1963, 1987, or 2010 are due to a large extent to the atmospheric
circulation. The dynamic component of compound events shows no significant trend over the 1951-2022
period. This suggests that the large-scale circulation has not played an important role in the observed decrease
in the occurrence of compound events. Interestingly, the dynamic component of cold days substantially
decreases (-0.47 days per winter; Table 3), although the p-value does not reach the 0.05 significance level (p-
value=0.08). Large-scale circulation may therefore have contributed to more than 50% of the decline in cold
days occurrence (-0.87 days per winter, Table 3) observed between 1951 and 2022, suggesting that
anthropogenic forcing may not be the only driver of this trend. Finally, there is no significant trend in the
dynamic component of low wind days.

**4. Discussion and conclusions**
In the context of the energy transition, compound low wind and cold events could present a stronger
threat for the adequacy between the demand and supply in electricity in France. Therefore, it is crucial to
characterize these climate compound events and to better understand how their frequency has changed in the
past to better anticipate how they could evolve in the coming decades.

Compound low wind and cold events are defined with ERA5 data over the 1950-2022 period using a
wind capacity factor index, and a temperature index that captures the current sensitivity of the electricity
demand in France to temperature. As compound low wind and cold events mainly occur between November
and March, our analyses focus on this period.

Compound events are quite rare (2 days per winter on average), with a peak occurrence in January.
They are generally short-lived, with a mean duration of 2 consecutive days although they can last up to 13



consecutive days. There are large interannual differences in the number of compound events, from 0 to 13
days per winter.
Over the observational record, we find a statistically significant decrease in compound events
frequency (-0.19 day per decade) that tends to have amplified over the last four decades. This decrease is likely
driven by the significant negative trend in cold days, while the frequency in low wind days shows no
significant trend. Overall, these results suggest a decrease in climate-related risks for the adequacy between
electricity demand and supply related to compound low wind and cold events over the observed period,
considering the current electricity system.
The role of the atmospheric circulation in the occurrence of compound events is assessed using a set
of four weather types derived with the unsupervised k-means classification technique applied to low wind
days. The frequency of compound events in each weather type ranges from 4% to 12%. This reveals a diversity
of regional atmospheric circulations that can lead to the occurrence of compound events in France. The
weather type associated with the highest compound events frequency (WT3) presents pronounced positive
sea-level pressure anomalies over Iceland and negative anomalies west of Portugal. This weather type leads
to negative anomalies of wind speed and temperature throughout Europe, which might pose challenges to the
electricity system on a larger scale than just in France.
Overall, we find that the regional atmospheric circulation contributes significantly to the occurrence
of compound events and explains an important part of their interannual variability. Interestingly, the regional
atmospheric circulation shows no significant contribution in the observed decrease in compound events over
the 1951-2022 period in ERA5. In the case of cold days, however, the large-scale circulation might contribute
to approximately 50% of their observed decrease. Similarly, Deser and Phillips (2023) found that large-scale
circulation contributes to a third of the mean winter temperature trend in Europe over the last decades. As
large-scale circulation variability is likely to be largely internal in origin, this result may have implications for
near-term projections.

In this study, compound low wind and cold events are identified using a straightforward approach that
consists of identifying cold days and low wind days independently. This has the advantage of allowing the
assessment of the relative contribution of cold days and low wind days to the decrease in compound events.
Another approach consists in identifying compound events as days with high residual load (i.e., electricity
demand minus wind power production), i.e., days that need important availability of other power sources than
wind power, such as hydro-electricity or nuclear generation (Bloomfield et al., 2020a). Such approach could
help to test the sensitivity of compound events to different power system scenarios (e.g., with different wind
power installed capacity).
With the anticipated rapid increase in onshore and offshore wind farms, the impact of low wind
conditions on French electricity production is expected to increase. Conversely, the impact of cold events on
French electricity demand is expected to decrease due to climate change. The question of how the risk on the





adequacy between electricity production and demand associated with compound events will evolve in the next few decades is therefore multifaceted, depending on both the future level of installed wind power capacity and climate change. We plan to address this question in future work using climate projections from the latest Couple Model Intercomparison Project Phase 6.

Future risks for the electricity system will also depend on the amount of electricity that can be stored to modulate the variability of renewable energy production. In this context, long-lasting compound low wind and cold events at the European scale will be of particular relevance. The study of such long events impacting a large domain requires a large sample. The use of the ERA5 reanalysis in this context is therefore not appropriate. An interesting option is to use state of the art Earth System Models, which provide large ensembles of simulations that enable identifying a higher number of long and high impact compound events (Bevacqua et al., 2023).

How the occurrence of compound events will continue to evolve in a changing climate is also a crucial question in the context of the energy transition. This study lays a methodological groundwork for addressing this question. It can also serve as a reference for the evaluation and selection of climate models that could then be used to assess the projections in compound events. In particular, our findings highlight the important role of the regional atmospheric circulation in driving compound low wind and cold events in winter in France, and this contribution is therefore a relevant metric for model evaluation in this context.

## Statements & Declarations

**Fundings.** This study is part of a PhD project funded by Réseau de Transport d'Electricité (RTE).

**Competing Interests.** The authors declare they have no conflict of interest.

**Author contributions.** All authors contributed to the study conception and design. Data collection and analysis were performed by FC, MB and JB. All authors contributed to the interpretation of the results. The first draft of the manuscript was written by FC, MB and JB and all authors commented on previous versions of the manuscript. All authors read and approved the final manuscript.

**Data availability.** The ERA5 reanalysis data is available on the Copernicus Data Store (CDS) at https://cds.climate.copernicus.eu/ cdsapp#!/dataset/reanalysis-era5-single-levels?tab=overview (Hersbach et al., 2020). The MERRA-2 reanalysis data is available from NASA at https://disc.gsfc.nasa.gov/datasets/M2T1NXLND_5.12.4/summary (Gelaro et al., 2017). The E-OBS gridded in situ observation datasets is provided by the European Climate Assessment & Dataset and available at: https://www.ecad.eu/download/ensembles/download.php.



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
