# Peer review of "Compound winter low wind and cold events impacting the French electricity system: observed evolution and role of large-scale circulation"

_EGUsphere, 2024_

## Author Comment (AC1)

**Compound winter low wind and cold events impacting the French electricity system: observed evolution and role of large-scale circulation**
Response to reviewers

We would like to thank both reviewers for their helpful comments and suggestions, which greatly helped to improve the manuscript. Their comments are shown below in black, with our responses in blue. In response to the major and minor points raised by both reviewers, we have added a Supplementary Materials document to the manuscript. Additionally, we made some aesthetic adjustments to the figures to improve their clarity.

**Reply to reviewer 1:**

Line 39. Onshore wind power capacity will increase to 30-39GW also by 2035? Please specify it.

Yes, wind power capacity is planned to increase to 30-39GW by 2035 according to RTE's scenarios (RTE, 2023). We clarified this in the manuscript at L.42-44:
"Onshore wind power capacity is planned to increase from 20GW in 2022 to 30-39GW by 2035 and substantial additional offshore wind farms are also planned, with a total projected capacity of 18GW by 2035 compared to 0.5GW in 2022 (RTE, 2023)."

Line 110. Is MERRA interpolated to ERA5 resolution of 0.25º?

Thanks for raising this question. MERRA-2 was not interpolated on the ERA5 grid before the calculation of our indices. Reviewer 2 also raised some comments regarding the interpolation scheme (nearest neighbor vs. bilinear) used to calculate our indices. In order to address these comments, we have conducted a series of sensitivity tests. These tests show that our indices are not affected by the interpolation scheme. Therefore, interpolating MERRA-2 to ERA5 should not affect our results. Here are our responses to the comments from Reviewer 2:

"To test whether the interpolation method has an influence on our indices, a bilinear interpolation scheme was used to interpolate 100-m wind speed at wind turbine locations, and near-surface temperature at each city coordinates, for ERA5, MERRA-2 and E-OBS datasets. As the reviewer suggests, the near-surface temperature is adjusted to each station altitude prior to the bilinear interpolation for the calculation of the temperature index. The results were compared to those obtained using the nearest neighbor interpolation, as described in the article. The figure below shows a general good match between the results obtained with the two interpolation methods, for the temperature index in ERA5, MERRA-2 and E-OBS, as well as the wind capacity factor index for ERA5 and MERRA-2. This suggests that the interpolation method has only a minimal impact on the findings of this study.

[Figure]

Figure R2.1: Temperature index as calculated with near-surface temperature interpolated using the nearest neighbor method (X-axis) versus the bilinear interpolation (Y-axis) at each station for (a) ERA5 (b) MERRA-2 and (c) E-OBS datasets for the winters of the 1950-2022 period. Wind capacity factor index as calculated with wind speed interpolated with the nearest method (X-axis) against as calculated with bilinear interpolation (Y-axis) at each wind farm for (d) ERA5 and (f) MERRA-2 datasets for the winters of the 1950-2022 period. The mean difference and the correlation coefficient between the two interpolation methods are shown in the top left corner."

Lines 162-164: Figures 1 and 2. Why not to have in the same plot wind and temperature in another plot? E.g., figure 1 a) with 2 b).

We appreciate this suggestion and have implemented the recommended modifications. The revised figures are included below:

[Figure]

Figure 1: (a) Spatial distribution of the wind power installed capacity (MW) in France in 2021 from the WindPower.net dataset used for the calculation of the wind capacity factor index. (b) National French wind capacity factor index as calculated with ERA5 (no unit; X-axis) versus observations (no unit; Y-axis) in winter over the 2012-2020 period. The correlation coefficient is given in the top left corner, and the black dashed line represents the y:x function

[Figure]

Figure 2: (a) Location of the 32 French cities and associated weights (no unit) used for the calculation of the temperature index. (b) Temperature index as calculated in ERA5 (°C; X-axis) versus observations of the electricity demand (GW; Y-axis) in winter over the 2012-2020 period, excluding week-ends and bank holidays. The correlation coefficient is given in the top right corner. The linear regression line between the temperature index and the electricity demand observations is shown by the black dashed line. The corresponding linear regression equation, in the form y=y(15°C)+a*(15°C-x), where 15°C is the threshold of residential heating and a the thermosensitivity of the electricity demand, is shown in the top right corner.

Line 180. Can the authors explain more why low wind are based on the 23th percentile? Is there any reasoning behind? I think this point is important.

We acknowledge that the explanation of our choice of the threshold needed some clarifications, and we thank the reviewer for raising this issue.

Low wind days are defined as days with an observed wind capacity factor below 0.15 (in the éCO2mix dataset, section 2.2). This value corresponds to the 23th percentile of the distribution of wind capacity factor in winter in the observations. We acknowledge that highlighting the 23th percentile threshold in the former version of the manuscript may have appeared somewhat unconventional and cherry-picked, which is less the case for the threshold 0.15. Therefore, we revised the section 2.5 "Identification of low wind days, cold days and associated compound events" at L.216-218:

"Days of low wind capacity factor (red points in Figure 3) are defined as days with an observed wind capacity factor below 0.15, corresponding to the 23th percentile of its distribution in winter."

Additionally, we would like to inform the reviewer that, in response to comments from Reviewer 2, we have included sensitivity tests regarding the definition of compound events in the Supplementary Materials. In these tests, two alternative definitions are used and compared with the main definition. The first one tests a more extreme threshold for the wind capacity factor (i.e., 5th percentile) index compared to the temperature index (i.e., 23th percentile). The second one tests identical thresholds for both indices to define compound events (i.e., 10th percentile for both indices). These tests show limited sensitivity to thresholds for the definition of compound events, except for the long-term trend in the observed occurrence of compound events over the 1951-2022 period.

Line 263. Figure 5. Why there is not green line (E-OBS) in 5a?

There are no green lines in Figure 5a, and also Figure 5c because we only use the near-surface temperature variable from the E-OBS dataset. We chose not to use E-OBS for calculating the wind capacity factor index because (1) the E-OBS wind speed dataset only starts in 1980 and (2) the spatial coverage of the source stations is not good, especially in the early period, which is problematic for our application. Therefore, low wind days and compound events are not shown in Figure 5a and 5c for the E-OBS dataset.

Line 305. This can be explained as the compound seem to be mostly driven by cold temperatures, so their patterns are very similar.

We agree with the reviewer that the higher similarities in the composite of mean sea-level pressure anomalies between cold days and compound events (Figure 6a, b), compared to low wind days and compound events (Figure 6a, c) can be explained by the stronger link between cold days and compound events, as noted in their climatological characteristics (Figure 4) or temporal evolution (Figure 5).

This stronger link is due to the compound event definition, which is based on a more extreme threshold for cold days compared to low wind days. In response to comments from Reviewer 2 concerning the definition of compound events, we explored an additional compound event definition based on a more extreme threshold on the wind capacity factor index. While we found that the associated composite of mean sea-level pressure anomalies (red contours of Figure S3) is somewhat more similar to the composite of mean sea-level pressure anomalies for low wind days (Figure 6c), the main conclusions of this work are generally not sensitive to these thresholds. We clarified this point by adding at (L. 433-439):

"We find relatively higher similarities in the mean sea-level pressure anomalies between cold days and compound events compared to between low wind days and compound events. This can be explained by a more extreme threshold used for cold days compared to low wind days in the definition of compound events. Note that the sensitivity to thresholds used in the definition of compound events is documented in Supplementary Materials. While we find that sea-level pressure anomalies between low wind days and compound events compare better when setting a more extreme threshold for low wind days in the compound event definition, the main conclusions of this work are generally not sensitive to these thresholds (Figure S3)."

Line 403. "Overall" is repeated twice in the same sentence.

Thanks, it is corrected (L.560): "This leads to a weak decrease of 20% in the frequency of compound events."

Lines 406-407. This is not very clear, can you please clarify what do you mean with "a simple change in the frequency"?

We acknowledge that the term "simple change" was misleading and have clarified this in the revised manuscript by modifying the sentence at (L.563): "However, due to significant intra-type variability, a change in the frequency of a limited number of weather types may not capture the full range of circulation changes."

Lines 429-436. The authors state that the large-scale atmospheric circulation did not have influence in the observed decreased in the occurrence compound events, while it did in the occurrence of cold days. But, if the compound events are mostly driven by the cold days, how can the authors explain this?

Thanks for raising this question. Following a comment from reviewer 2, we performed some sensitivity tests to the parameters used in the dynamic adjustment analysis. We either found a significant decrease in the evolution of circulation-induced compound events or no trend depending on the choice of the parameters. The main conclusion from these sensitivity tests is that it is rather difficult to conclude whether or not the large-scale circulation played a role in the observed decrease in compound events because of limited robustness.

Following these sensitivity tests, we revised our choice of parameters in the dynamical adjustment analysis. With the revised parameters, the circulation-induced compound events exhibit a significant decrease (-0.14 days per decade, p=0.04). However, we decided to remain cautious in our conclusions as, as said in the previous paragraph, the significance of the trend of circulation-induced compound events is not robust to the methodology. Here are our revisions:

- In the abstract section at L. 25-28: "We further show that the atmospheric circulation and its internal variability are likely to play a role in the observed reduction in cold days, suggesting that this negative trend may not be entirely driven by anthropogenic forcings. It is however more difficult to conclude on the role of the atmospheric circulation in the observed decrease in compound events."
- Update of the revised dynamical adjustment parameters in the methodological section 2.7.

- In section 3.2 at L.593-601: "Interestingly, circulation-induced cold days substantially decrease (-0.40 days per decade; Table 3), although the p-value does not reach the 0.05 significance level (p-value=0.14). Large-scale circulation may therefore have contributed to more than 50% of the decline in cold days occurrence (-0.72 days per decade, Table 3) observed between 1951 and 2022, suggesting that anthropogenic forcing may not be the only driver of this trend. Similarly, circulation-induced compound events show a decrease (-0.14 days per decade, Table 3) over the 1951-2022 period (p-value=0.04). However, both the trend significance and the magnitude of the slope are sensitive to the parameters used in the dynamical adjustment (not shown). Thus, the robustness is too weak and prevents us from drawing conclusions on the role of the large-scale circulation on the decrease in compound events. Finally, there is no significant trend in the circulation-induced low wind days."
- In the conclusion section at L.677-680: "Interestingly, the large-scale atmospheric circulation shows a contribution of approximately 50% of the observed decrease in cold days over the 1951-2022 period in ERA5. [...] Finally, we cannot conclude on the role of large-scale circulation in the decrease of compound events as our methodology exhibits sensitivity to its parameters."

|  | Compound events | Cold days | Low wind days |
|---|---|---|---|
| ERA5 | **-0.19 (0.02)** | **-0.72 (0.02)** | -0.08 (0.59) |
| Circulation-induced in ERA5 | **-0.14 (0.04)** | -0.40 (0.14) | -0.21 (0.75) |

Table 3: (first row) Trend (slope in days/decade, and associated p-value) in the frequency of low wind days, cold days, and compound low wind and cold events in winter over the period 1951-2022 in ERA5 (first row; same trend estimates as in Table 1) and in their respective circulation-induced events (second row; section 2.7). The slope is calculated with the Theil-Sen estimator and the p-value is calculated with the Mann-Kendall test. Significant trends with p<0.05 are shown in bold.

Line 434. Where is -0.87 in table? This must be a mistake, please correct it.

As described in our reply to the previous comment, please note that we have made some changes to the dynamic adjustment parameters following a comment from reviewer 2. We changed Table 3 accordingly in the main text. The revised Table 3 is included in the response to the previous comment.

Line 473-474. Can you be more specific?

We clarified this sentence and revised the main text at L. 681-686 :

"Assuming that observed changes in the large-scale circulation are mainly driven by internal climate variability (Shepherd, 2014), these results suggest that, over the last few decades, climate variability likely reinforced the long-term decline in cold events in response to warming. This may not continue in the near future, potentially leading to a temporary increase in the occurrence of cold events."

Lines 487- 490. I would also add to this multifaceted problem the changes in the demand patterns and therefore, the changes in the compound events. Here, compound events are

limited to cold days and low production. But if the demand increases in summer due to higher temperatures, this variability in compound events would change as well.

Thanks, this suggestion was also raised by Reviewer 2. We have added an additional discussion in the conclusion about future changes in demand patterns. Here is the section available at (L. 696-708):

"With the anticipated rapid growth of onshore and offshore wind farms, the impact of low wind conditions on power system risks is likely to increase and to become a greater threat alongside cold temperature conditions. As climate change reduces the frequency of cold events (Seneviratne, 2021), future risks to the French power system may be more evenly spread throughout the winter season, rather than being concentrated primarily in January and February as it is currently (RTE, 2023, §6.2.5.3). In addition, changes in electricity demand patterns are also anticipated. During summer, increased electricity demand is expected due to higher use of air conditioning in France. However, the risks for the French power system during summer are expected to be limited thanks to higher solar power production and power system flexibilities (RTE, 2023, §6.2.5.3). How the risk on the adequacy between electricity generation and demand associated with compound events will evolve in the next few decades is therefore multifaceted, depending on future levels of installed wind power capacity, changes in demand patterns, and climate change. We plan to address some of these questions in future work using climate projections from the latest Couple Model Intercomparison Project Phase 6."

**References**

Añel, J., Fernández-González, M., Labandeira, X., López-Otero, X., and De La Torre, L.: Impact of Cold Waves and Heat Waves on the Energy Production Sector, Atmosphere, 8, 209, https://doi.org/10.3390/atmos8110209, 2017.

RTE (Réseau de transport d'électricité). Bilan prévisionnel, édition 2023. Futurs énergétiques 2050. 2023-2035 : première étape vers la neutralité carbone.

RTE (Réseau de transport d'électricité), Futurs énergétiques 2050. Les scénarios de mix de production à l'étude permettant d'atteindre la neutralité carbone à l'horizon 2050, octobre 2021.

Shepherd, T. G.: Atmospheric circulation as a source of uncertainty in climate change projections, Nature Geosci, 7, 703–708, https://doi.org/10.1038/ngeo2253, 2014.

---

## Author Comment (AC2)

**Compound winter low wind and cold events impacting the French electricity system:**
**observed evolution and role of large-scale circulation**
Response to reviewers

We would like to thank both reviewers for their helpful comments and suggestions, which greatly helped to improve the manuscript. Their comments are shown below in black, with our responses in blue. In response to the major and minor points raised by both reviewers, we have added a Supplementary Materials document to the manuscript. Additionally, we made some aesthetic adjustments to the figures to improve their clarity.

**Reply to reviewer 2:**

The description of the study is confusing in suggesting that the compound events are defined on the meteorological variables and not on electricity production, but the authors are using wind capacity factors, so they are not necessarily identifying low wind events, but rather low wind generation days. How are the authors ensuring that the so called 'low wind' conditions are not due to high wind speeds above cut-off?

We acknowledge that there was some confusing part in the previous version of our manuscript regarding the definition of compound low wind and cold events. This has been clarified.

Indeed, we identify compound low wind and cold events using a wind capacity factor index and a temperature index. These indices were defined to be relevant for the French power system, compared to using raw wind speed and temperature data (e.g. averaged over France). We agree with this reviewer that theoretically both low or high wind speed values can cause low values of the capacity factor index. However, our sample of low wind days only includes days with rather low mean wind speeds at 100m in France, as shown below with the figure R2.1. While we find some hours associated with high wind speed values and low wind capacity factor values during the Lothar storm occurring in December 1999 in France, we find that this does not happen at the daily timescale.

We clarified the description of compound low wind and cold events definition in the manuscript by revising section "2.5 Identification of low-wind days, cold days and associated compound events" at L. 215-221:

"In this study, compound events are defined as days when low wind capacity factor and cold temperature co-occur (green points in Figure 3). Days of low wind capacity factor (red points in Figure 3) are defined as days with an observed wind capacity factor below 0.15, corresponding to the 23th percentile of its distribution in winter. This sample of low wind capacity factor days only captures days with low values of 100-m wind speed over France (see Figure S1). Thus, these events are referred to as low wind days. Cold days are defined as days with the temperature index below 0°C, corresponding to the 5th percentile of its distribution in winter (blue points in Figure 3)."

Accordingly, we included the figure R2.1 in the Supplementary Materials, as Figure S1:

[Figure]

Figure S1: Wind capacity factor index (Y-axis) against 100-m wind speed wind averaged over France (X-axis) as calculated with ERA5 for each day of the 1950-2022 period.

We have also added a paragraph at the beginning of the section "2 Data and Method" to clarify earlier in the manuscript that the definition of compound low wind and cold events is based on indices that are relevant for the French electricity demand and wind power production. See at L. 101-107:

"In this study, we identify compound low wind and cold events based on a wind capacity factor index and a temperature index. These indices respectively capture the sensitivity of the French wind power production to wind speed conditions and the sensitivity of the French electricity demand to temperature conditions. Thus, compound events as defined in this study correspond to days when the French power system is challenged by both wind and temperature conditions. In this section, we first introduce the data and methodology used to define the wind capacity factor and temperature indices and compound events. Then, we introduce the methodology used to identify compound low wind and cold events. Finally, methodologies used to identify weather types and to assess the role of the atmospheric circulation in the evolution of compound events are developed."

There is an asymmetry in the definition of the extreme events for the single variables (in terms of the percentile thresholds) that is not discussed. Relaxing the definition of the temperature threshold could have led to a larger sample of compound events and less sensitivity of the results for compounds to cold days that is seen throughout the analysis.

We thank the reviewer for raising this point. We analyzed the sensitivity of compound events properties to wind capacity factor and temperature thresholds. We explored additional compound event definitions and concluded that setting a more extreme threshold on either the wind capacity factor or the temperature index in the definition of compound events has limited impacts on the characteristics of compound events over the 1950-2022 period, with the exception of the long-term trend significance for cold days. We have included these analyses in Supplementary Materials:

**"Sensitivity of compound events to their definition**

In this section, the sensitivity of compound events properties to the thresholds applied to the wind capacity factor and temperature indices is discussed. In this study, compound events

are defined based on a more extreme threshold for the temperature index (i.e., 5th quantile, or 0°C in ERA5) compared to the wind capacity factor index (i.e., 23th quantile, or 0.11 in ERA5). Here, two alternative definitions are used and compared with the main definition. The first one tests a more extreme threshold for the wind capacity factor index (i.e., 5th quantile, or 0.045 in ERA5) compared to the temperature index (i.e., 23th quantile, or 3.6°C in ERA5). The second one tests identical thresholds for both indices to define compound events (i.e., 10th quantile for both indices, or 0.065 and 1.5 °C in ERA5, respectively).

Overall, we find that the climatological characteristics of compound events remain generally similar between these different definitions, with a comparable number of compound events that are similarly distributed across the extended winter months and have similar durations (Figure S2).

[Figure]

Figure S2: (a) Monthly mean number of compound low wind and cold events;(b) distributions of the number of compound events per winter; (c) distribution of the duration (in days) of compound events in winter over the 1951-2022 period in ERA5. Colors refer to the different definition tested for the identification of compound low wind and cold events: (green) identical thresholds for both the temperature and wind capacity factor indices, (blue) more extreme threshold for the temperature index compared to the wind capacity index as used in the main text (5th vs 23rd percentile, respectively), and (red) more extreme threshold for the wind capacity factor index compared to the temperature index (5th vs 23rd percentile, respectively). The solid line and the dashed line in the boxplots in (b) and (c) show the median and the average, respectively.

Further exploring the sensitivity to the definition on the observed evolution, we find that, while compound events defined with a more extreme temperature threshold significantly decrease during the 1951-2022 period in ERA5 (p=0.02, Figure 5 and Table 1 of the main text), the other two definitions of compound events tested here do not exhibit a significant trend over this period (p=0.38 and p=0.28, respectively; Table S2). For these two alternative definitions, the absence of trend might result from the lower influence of cold days and the higher influence of low wind days in the interannual evolution of compound events (Table S1), as cold days exhibit a significant decrease while low wind days exhibit an absence of trend (Table S2).

| Compound event definition | Pearson correlation | r(low wind days, compound events) | r(cold days, compound events) |
|---|---|---|---|
| | | | |

| | 0.20 | 0.85 |
|---|---|---|
| More extreme temperature threshold | 0.20 | 0.85 |
| Identical threshold | 0.40 | 0.67 |
| More extreme wind capacity factor threshold | 0.51 | 0.41 |

Table S1: Values of the Pearson correlation coefficient between the interannual evolution of low wind days and compound events (first column) and cold days and compound events (second column) for each compound event definition (per raw) over the 1950-2022 period, in ERA5.

| Compound event definition | Long-term trend Low wind days | Cold days | Compound events |
|---|---|---|---|
| More extreme temperature threshold | -0.08 (0.59) | -0.72 (0.02) | -0.19 (0.02) |
| Identical threshold | -0.38 (0.27) | -1.5 (0.00) | 0.0 (0.28) |
| More extreme wind capacity factor threshold | -0.28 (0.16) | -2.5 (0.00) | 0.0 (0.38) |

Table S2: Trend (slope in days/decade) and associated p-value in the number of compound low wind and cold events, cold days, and low wind days for each compound event definition over the 1950-2022 period in ERA5. The slope is calculated using the Theil-Sen estimator and the p-value with the Mann-Kendall test. Significant trends (p-value<0.05) are highlighted with a grey shaded cell.

We then investigate the average large-scale circulation pattern associated with the different definitions of compound events. Overall, we find that the three definitions lead to similar features: strong positive mean sea-level pressure anomalies over the British Isles and relatively less intense negative anomalies centered over the Azores (Figure S3b). The location of the positive sea-level pressure anomalies over the British Isles slightly varies according to compound event definition. We then investigate whether these average differences are linked with differences in weather type frequencies (see section 2.6 and 3.2). For that, we project compound events onto the low wind day weather types used in section 2.6., which is made possible by the fact that compound events, as they are defined, are included in the subset of low wind days. The weather type WT3 is the most frequent for all compound event definitions, despite a lower frequency in compound events defined with identical and more extreme wind thresholds (Figure S3c).

Overall, this sensitivity analysis shows that setting a more extreme threshold on either the wind capacity factor or the temperature index in the definition of compound events has limited impacts on the characteristics of compound events over the 1951-2022 period, except for the observed evolution and trend estimate.

[Figure]

Figure S3: (a) Interannual evolution of the number of compound events per winter, with dashed lines showing the linear trend (calculated with the Theil-Sen estimator), (b) composite of sea-level pressure, with solid contours corresponding to positive anomalies, (c) frequency of compound events for each weather type (in % of weather type size). Colors correspond to the different definitions of compound events tested: (green) identical thresholds for both the temperature and wind capacity factor indices, (blue) more extreme threshold for the temperature index compared to the wind capacity index as used in the main text (5th vs 23rd percentile, respectively), and (red) more extreme threshold for the wind capacity factor index compared to the temperature index (5th vs 23rd percentile, respectively)."

In this study, we have taken a bottom-up approach and therefore chosen to define compound events with thresholds on temperature and wind capacity factor index that are relevant for the power system. We chose to set a more extreme threshold for the temperature index compared to the wind capacity factor index as cold events have historically been related to risks to the French power system (Añel, 2017). We made some clarifications in the section 2.5 "Identification of low-wind days, cold days and associated compound events" by adding at (L. 221-228):

"In this study, we chose to set a more extreme threshold for the temperature index compared to the wind capacity factor index because risks to the French power system have historically been primarily related to the occurrence of cold waves in winter (Añel, 2017). However, depending on future levels of wind power installed capacity and demand patterns, the sensitivity of the power system to these thresholds might change. Sensitivity tests exploring different thresholds for both indices are therefore included in Supplementary Materials. These tests show limited sensitivity to thresholds for the definition of compound

events, except for the long-term trend in the observed occurrence of compound events over the 1951-2022 period."

The study is missing a comparison with other work considering weather types, copula approaches or circulation types with similar purposes. Though these body of research is presented in the introduction, there is no comparison or discussion. What is this work adding to the current understanding of regional circulation relevance of compound extremes?

Following this comment, we have added a comparison of our results with other studies in the conclusion section at (L. 659-671):
"Other studies focusing on compound low wind and cold events at the scale of Europe also highlight the role of large-scale circulation in compound event occurrence. Bloomfield (2019) and Tedesco (2023) find that pronounced positive mean sea-level pressure anomalies over Northern Europe and negative anomalies over the Azores lead to a large number of compound events in Central and Western Europe, and this circulation pattern projects well onto the weather type WT3 of this study. Similarly, Otero (2022) finds that a particular weather type (called Greenland blocking), which is similar to our weather type WT3, increases the probability of compound events in Europe. This is also true for a second weather type (called European blocking) that projects relatively well onto our weather type WT2. Hence, in this study, we identify large-scale circulation patterns associated with compound events in France that compare broadly with previous findings focused over Europe. There are slight discrepancies in the location of the positive and/or negative anomalies, and these might be partly explained by differences in the particular domain of interest. Other methodological differences such as weather types calculation or definition of compound events might also explain some differences."

To avoid repetitions in the manuscript between the modified Conclusions section and the Introduction, we simplified the section in the introduction that discusses the influence of large-scale circulation on the occurrence of compound low wind and cold events by removing L. 90:
"Tedesco et al. (2023) showed that compound low wind and cold events in France are mostly associated with positive anomalies of geopotential height at 500hPa over Iceland and negative anomalies over the Azores. Otero et al. (2022b) showed that situations of limited production of electricity from wind and solar energies co-occurring with cold events are mostly associated with positive anomalies of geopotential height at 500hPa over the North Sea region."

Line 116, page 4: should be 'and so on'.

Thank you, it is corrected in the manuscript at L 133-134: "By convention, hereafter, winter 1951 corresponds to the period from November 1950 to February 1951 and so on."

Lines 151-157, page 5: what is the impact of using the closest grid point to each site rather than for example a bilinear interpolation? Was this tested? It could reduce the dependence on the resolution of the gridded dataset, which is key when comparing ERA5 and MERRA-2, and the height-scaling could be applied before the interpolation. Also, it is not described how hourly wind speeds are obtained for each wind farm site. Is the same approach used?

We thank the reviewer for pointing out that the interpolation method used to obtain the 100-m wind speeds at wind farm sites was not described in the manuscript. A nearest neighbor interpolation scheme is used. We revised the description of the methodology to calculate wind capacity factor at L. (171-173):

"To calculate the wind capacity factor, ERA5 hourly wind speeds at 100 m are first interpolated to each wind farm site using a nearest neighbor interpolation scheme. The wind speeds are then extrapolated at hub height using a power law ($\alpha$=0.14; Manwell, 2010; van der Wiel et al., 2019a)."

Regarding the spatial resolution of the different datasets, there is indeed a difference in resolution between ERA5 (lat:0°25, lon:0°25) and MERRA-2 (lat:0.5°, lon:0.6°), while E-OBS has the same resolution as ERA5 (but with a slight offset). To test whether the interpolation method has an influence on our indices, a bilinear interpolation scheme was used to interpolate 100-m wind speed at wind turbine locations, and near-surface temperature at each city coordinates, for ERA5, MERRA-2 and E-OBS datasets. As the reviewer suggests, the near-surface temperature is adjusted to each station altitude prior to the bilinear interpolation for the calculation of the temperature index. The results were compared to those obtained using the nearest neighbor interpolation, as described in the article. The figure below shows a general good match between the results obtained with the two interpolation methods, for the temperature index in ERA5, MERRA-2 and E-OBS, as well as the wind capacity factor index for ERA5 and MERRA-2. This suggests that the interpolation method has only a minimal impact on the findings of this study.

[Figure]

Figure R2.1: Temperature index as calculated with near-surface temperature interpolated using the nearest neighbor method (X-axis) versus the bilinear interpolation (Y-axis) at each station for (a) ERA5 (b) MERRA-2 and (c) E-OBS datasets for the winters of the 1950-2022

period. Wind capacity factor index as calculated with wind speed interpolated with the nearest method (X-axis) against as calculated with bilinear interpolation (Y-axis) at each wind farm for (d) ERA5 and (f) MERRA-2 datasets for the winters of the 1950-2022 period. The mean difference and the correlation coefficient between the two interpolation methods are shown in the top left corner.

Lines 194-195, page 8: Looking at the distribution of cold days across weather types at a later stage does not imply accounting for the distribution of compound events. This needs to be better explained.

We acknowledge that there were some confusing parts in the previous version of our manuscript regarding the identification of the weather types, and how we ultimately account for compound events within these weather types. We have now carefully revised the manuscript to better explain the methodology (L. 263-266), as well as the relative interpretation of the results (L. 452-455).

(L. 263-266): "In other words, the weather types represent clusters of low wind days with similar large-scale circulation patterns. In a second phase, we examine how cold days are distributed across these different weather types. Finally, we can thus assess the number of compound event days for each identified weather type."

(L. 452-455).): "Four weather types are obtained by classifying the mean sea-level pressure during low wind days using the k-means algorithm (see section 2.6). We then assess the distribution of compound low wind and cold events across these four weather types to identify the most favorable synoptic situations leading to the occurrence of these compound events in France, and over western Europe more generally."

Lines 217- 220, page 8: How was the choice of 50 constructed analogues justified. For each day the authors are choosing 1500 analogues from a pool of ~11000, so you are forcing more than 10% of the days to be 'analogues' that seems like a stretch, then buy repeating the procedure 50 times, the chances of all analogues being extremely similar is really big. Some sensitivity testing must have been performed for the choice of these numbers?

We thank the reviewer for raising this topic.

Before proceeding with this response, please be aware that we have updated the terminology related to the dynamical adjustment method. What was previously referred to as the event's dynamical component in the manuscript is now called circulation-induced events. For further details, please refer to the next response.

We tested the sensitivity of our results to 2 parameters of our dynamical adjustment method (section 2.7 in the manuscript), which are the total number of "analogues" (referred to as N hereafter) and the number of analogues randomly selected from the pool of N analogues (referred to as Ns hereafter). First, we tested two values for N, namely N = 400 and N = 1500, the former being used in Terray, 2021 and the latter being the value previously used in the manuscript. Then, for each choice of N, we tested a range of Ns values: [100, 200, 300] for N=400, and [700, 1000, 1300] for N = 1500 (previously Ns=1300 in the manuscript). In addition, we increased the number of random selection iterations (referred to as Nr hereafter) from previously 50 in the manuscript to 200. The choice of 200 is motivated by a trade-off between having a higher number of realizations, which improves the estimation of the

indices' dynamical component, and computational costs. In addition, two metrics are used to explore the sensitivity of these different parameters in our analysis:

1.  The explained variance ($R^2$) in both the wind capacity factor index and the temperature index by the dynamic component of the wind capacity factor and the temperature index, respectively.

2.  The slope and the significance (at the 0.05 level) of the trend in the number of circulation-induced cold days, low wind days, and compound events per winter over the 1951-2022 period, using the same methodology as in the manuscript (see section 2.7)

Overall, the explained variance shows some sensitivity to N and Ns, with values ranging from 0.59 to 0.72 for the temperature index and from 0.49 to 0.66 for the wind capacity factor index (Figure R2.1 and R2.1b). The explained variance slightly increases with N=1500 compared to N=400 for the temperature index (by ~ 0.05, Table R2.1a) while it exhibits a more pronounced decrease of ~ - 0.15 with N=1500 compared to N=400 for the wind capacity factor index (Table R2.1b). Consequently, N=400 is preferred to N=1500 in the subsequent analysis. Then, given the value N=400, $R^2$ decreases from Ns=200 to Ns=300 for both the wind capacity factor and the temperature index. Therefore, we chose not to consider (N=400, Ns=300). In summary, we decided to keep the parameters (N, Ns) of (400,100) and (400, 200).

[Figure]

Figure R2.1: Explained variance ($R^2$) of (a) the temperature index and (b) wind capacity factor index by the dynamic component of the temperature index and wind capacity factor index, respectively, for different values of parameters (Ns, N) of the dynamical adjustment.

Then, we assess the sensitivity of the long-term trend metric for the two couples of parameters (N, Ns) of (400,100) and (400,200):

*   For circulation-induced low wind days, the trend exhibits non significant p-values of 0.75 for either couple of parameters. This is generally in agreement with our conclusions that large-scale circulation does not play a role in the long-term evolution of low wind days.
*   For circulation-induced cold days, the slope of the long-term trend is -0.5 days / decade (p-value=0.07) for (N, Ns) = (400,100), and a slope of -0.4 days / decade (p-value=0.14) for (N, Ns) = (400,200). Similarly, these slopes of the trends are of similar magnitude compared to our previous finding (slope = - 0.47 days per decade,

Table 3 of the previous manuscript), as well as the p-values that are also not statistically significant (p = 0.08, Table 3 of the previous manuscript). Therefore, these tests support our conclusion that the large-scale circulation is likely to play a role in the decrease of cold days occurrence over the 1951-2022 period.

- For circulation-induced compound events, the long-term trend exhibits a null slope (p-value=0.21) for (N, Ns) = (400,100), which supports our previous conclusion that large-scale circulation is likely to not play a role in compound event decrease. Conversely, the long-term trend exhibits a slope of -0.14 days per decade (p-value=0.04) for (N, Ns) = (400,200), leading to opposite conclusions. Since the methodology is sensitive to the N and Ns parameters, we cannot conclude with confidence whether or not the large-scale circulation is likely to play a role in the observed decrease of compound event decrease.

Based on these conclusions, we decided to change the dynamical adjustment parameters to N=400, Ns=200 and Nr=200. With the revised parameters, the circulation-induced compound events exhibit a significant decrease (-0.14 days per decade, p=0.04). However, we decided to remain cautious in our conclusions as, as said in the previous paragraph, the significance of the trend of circulation-induced compound events is not robust to the methodology. Here are our revisions:

- In the abstract section at L. 25-28: "We further show that the atmospheric circulation and its internal variability are likely to play a role in the observed reduction in cold days, suggesting that this negative trend may not be entirely driven by anthropogenic forcings. It is however more difficult to conclude on the role of the atmospheric circulation in the observed decrease in compound events."
- Update of the revised dynamical adjustment parameters in the methodological section 2.7.
- In section 3.2 at L.593-601 "Interestingly, circulation-induced cold days substantially decrease (-0.40 days per winter; Table 3), although the p-value does not reach the 0.05 significance level (p-value=0.14). Large-scale circulation may therefore have contributed to more than 50% of the decline in cold days occurrence (-0.72 days per decade, Table 3) observed between 1951 and 2022, suggesting that anthropogenic forcing may not be the only driver of this trend. Similarly, circulation-induced compound events show a decrease (-0.14 days per decade, Table 3) over the 1951-2022 period (p-value=0.04). However, both the trend significance and the magnitude of the slope are sensitive to the parameters used in the dynamical adjustment (not shown). Thus, the robustness is too weak and prevents us from drawing conclusions on the role of the large-scale circulation on the decrease in compound events. Finally, there is no significant trend in the circulation-induced low wind days."
- In the conclusion section at L.677-680: "Interestingly, the large-scale atmospheric circulation shows a contribution of approximately 50% of the observed decrease in cold days over the 1951-2022 period in ERA5. [...] Finally, we cannot conclude on the role of large-scale circulation in the decrease of compound events as our methodology exhibits sensitivity to its parameters."

[Figure]

Figure 9: Interannual evolution of the number of (a) circulation-induced compound low wind and cold events, (b) cold days, and (c) low wind days in winter over the 1951-2022 period in ERA5. For each event, the value of the correlation coefficient between the inter-annual evolution and its respective circulation-induced evolution is shown in the upper left. Dashed lines show the linear trend (calculated using the Theil-Sen estimator; see Table 3 for the slope value and associated p-value).

|  | Compound events | Cold days | Low wind days |
|---|---|---|---|
| ERA5 | **-0.19 (0.02)** | **-0.72 (0.02)** | -0.08 (0.59) |
| Circulation-induced in ERA5 | **-0.14 (0.04)** | -0.40 (0.14) | -0.21 (0.75) |

Table 3: (first row) Trend (slope in days/decade, and associated p-value) in the frequency of low wind days, cold days, and compound low wind and cold events in winter over the period 1951-2022 in ERA5 (first row; same trend estimates as in Table 1) and in their respective circulation-induced events (second row; section 2.7). The slope is calculated with the Theil-Sen estimator and the p-value is calculated with the Mann-Kendall test. Significant trends with $p < 0.05$ are shown in bold.

Lines 237-238, page 9: "when both the dynamic component of low wind days and cold days occur". This phrase does not make a lot of sense. What does it mean that the dynamic component occurs? It is a construction produced by averaging a lot of different things, so it is not really something that 'occurs'.

We recognize that the previous version of our manuscript had some confusing elements concerning the terminology related to the dynamical adjustment method. To clarify the terminology, we revised the corresponding methodology section. Please note that we made careful revisions to the text throughout the entire manuscript.

(L. 317-323): "To isolate the impact of large-scale circulation on the evolution of compound events, we define circulation-induced compound events. These are virtual events based only on the contribution of large-scale circulation. First, circulation-induced low wind days and cold days are identified using the same thresholds as for the definition of low wind days and cold days (i.e., the 5th percentile and the 23rd percentile of the extended winter distribution, respectively; Section 2.5), but this time on the dynamic component of the wind capacity factor and temperature indices, respectively. Finally, circulation-induced compound events are identified as days when both the circulation-induced low wind days and circulation-induced cold days virtually occur."

Accordingly, we also changed "dynamic component" to "circulation-induced" in section 2.7, Figure 9 and Table 3, and section 3.2.

Figure 4: how is the seasonality of events presented over the full calendar year when they were defined for the extended winter only? Is the same threshold definition applied? How is that justified in terms of the percentiles? I think presenting the full year here is confusing and unnecessary.

Events were defined using the same thresholds over the whole year, which are computed with percentiles of the extended winter period distribution of indices. We agree that presenting the full year could be confusing. We now show the distribution of compound events within the extended winter period only in Figure 4a:

[Figure]

Figure 4: (a) Monthly mean number of compound low wind and cold events (green), cold days (blue), and low wind days (red); Distributions of (b) the number of days per winter and (c)

duration of compound low wind and cold events, cold days, and low wind days in winter over the 1951-2022 period in ERA5. The solid line and the dashed line in the boxplots in (b) and (c) show the median and the average, respectively.

Table I: It reads 'empty cells correspond to missing data', but there are no empty cells but rather '/'. Also, this actually means different things, for example an incomplete period in MERRA-2, but an unavailable variable in the case of E-OBS.

Thanks, it is corrected.

Lines 286-287, page 11: The larger sampling is a consequence of methodological choices, since a less extreme definition was considered for 'low wind' days.

We clarified that in the manuscript by replacing at (L. 383-385): "as defined in this study (section 2.5)" by "This is due to the more extreme threshold applied on the temperature index and therefore the larger sample of low wind days per winter on average compared to the number of cold days (section 2.5 and sensitivity analyses in Supplementary Material).".

Figures 6 and 7: they refer the wind composites to a % of climatological mean that is not clearly defined. Is this a seasonal mean? For the full extended winter? Is it the average of each day w.r.t. its daily climatological benchmark?

We clarified this point in the manuscript by adding at (L. 409-410):

"Relative anomalies for both the temperature and 100-m wind speed are calculated with respect to their daily climatology (1950-2022) in ERA5 (smoothed with a 15-day moving average)."

We also added a description of the method to calculate temperature index anomalies in the legend of Figure 8 at (L. 514-515):

"Temperature index anomalies are calculated with respect to the daily climatology (1950-2022) in ERA5 (smoothed with a 15-day moving average)."

Line 319, page 12: in the following what?

Thanks, we meant "in the following section". It is clarified in the new version of the manuscript at L.441 : "[..] further explored in the following section using a weather type analysis."

Line 423, page 18: is large-scale circulation a fair description? The domain used for the analogues would seem to constraint it to regional circulation?

We agree with the reviewer that it is somewhat ambiguous. We are not aware of an universally accepted definition of "large scales" for atmospheric circulation. The domain that we use for classification, whose limits are [30°W-30°E/33°S-70°N], covers a large part of the North-Atlantic and of Europe, and is large enough to include multiple weather systems such as extra-tropical cyclones or anticyclones occurring simultaneously. The pressure systems associated with the weather types cover hundreds of kilometers. We therefore think that it is

fair to talk about large-scale circulation although it would not be shocking to use "regional" either.

For the sake of consistency, we changed all instances of "regional" in the manuscript to "large-scale".

Lines 487-489, page 20: this discussion should include potential future changes in demand. For example, increases in warm seasons demand could lead to these events being more relevant in transition periods, as it was shown than "low wind" days are even more frequent then.

We thank the reviewer for this suggestion. We modified this paragraph to more comprehensively address the projected impact of changes in installed wind power capacity and demand patterns on power system risks at (L. 696-710):

"With the anticipated rapid growth of onshore and offshore wind farms, the impact of low wind conditions on power system risks is likely to increase and to become a greater threat alongside cold temperature conditions. As climate change reduces the frequency of cold events (Seneviratne, 2021), future risks to the French power system may be more evenly spread throughout the winter season, rather than being concentrated primarily in January and February as it is currently (RTE, 2023, §6.2.5.3). In addition, changes in electricity demand patterns are also anticipated. During summer, increased electricity demand is expected due to higher use of air conditioning in France. However, the risks for the French power system during summer are expected to be limited thanks to higher solar power production and power system flexibilities (RTE, 2023, §6.2.5.3). How the risk on the adequacy between electricity generation and demand associated with compound events will evolve in the next few decades is therefore multifaceted, depending on future levels of installed wind power capacity, changes in demand patterns, and climate change. We plan to address some of these questions in future work using climate projections from the latest Couple Model Intercomparison Project Phase 6."

**References**

Deser, C. and Phillips, A. S.: A range of outcomes: the combined effects of internal variability and anthropogenic forcing on regional climate trends over Europe, Nonlin. Processes Geophys., 30, 63–84, https://doi.org/10.5194/npg-30-63-2023, 2023.

RTE (Réseau de transport d'électricité). Bilan prévisionnel, édition 2023. Futurs énergétiques 2050. 2023-2035 : première étape vers la neutralité carbone.

RTE (Réseau de transport d'électricité), Futurs énergétiques 2050, octobre 2021.

Terray, L.: A dynamical adjustment perspective on extreme event attribution, Weather Clim. Dynam., 2, 971–989, https://doi.org/10.5194/wcd-2-971-2021, 2021.